# Resistant Protein: Forms and Functions

**DOI:** 10.3390/foods11182759

**Published:** 2022-09-08

**Authors:** Emanuele Zannini, Aylin W. Sahin, Elke K. Arendt

**Affiliations:** 1School of Food and Nutritional Sciences, University College Cork, College Road, T12 K8AF Cork, Ireland; 2APC Microbiome Institute, T12 K8AF Cork, Ireland

**Keywords:** resistant protein, protein digestibility, protein structure, food structure, food design

## Abstract

Several global health risks are related to our dietary lifestyle. As a consequence of the overconsumption of ultra-processed and highly digestible protein (150–200% of the recommended value), excess dietary proteins reach the colon, are hydrolysed to peptides and amino acids by bacterial proteases and fermented to various potentially toxic end products. A diet reformulation strategy with reduced protein content in food products appears to be the most effective approach. A potential approach to this challenge is to reduce food digestibility by introducing resistant protein into the diet that could positively influence human health and gut microbiome functionality. Resistant protein is a dietary constituent not hydrolysed by digestive enzymes or absorbed in the human small intestine. The chemical conformation and the amino acid composition strictly influence its structural stability and resistance to in vivo proteolysis and denaturation. Responding to the important gap in our knowledge regarding the digestibility performance of alternative proteins, we hypothesise that resistant proteins can beneficially alter food functionality via their role in improving metabolic properties and health benefits in human nutrition, similar to fibres and resistant starches. A multidisciplinary investigation of resistant protein will generate tremendous scientific impact for other interlinked societal, economic, technological and health and wellbeing aspects of human life.

## 1. Introduction

Food systems are under pressure to feed the world’s growing population within the planetary boundaries while ensuring the livelihoods of millions of people working along the food chain from farm to fork and the sector’s environmental sustainability [1]. With a projected population growth of 9.6 billion by 2050 and 10.6 billion by 2100, the global appetite for meat and animal products will increase by 76% by 2050 [1]. Addressing this will necessitate more sustainable production of protein sources for human and animal nutrition. Future food farming systems targeting microalgae, single-cell protein, insect larvae and cellular agriculture can secure the production of alternative food/protein sources in a closed environment with consistent and efficient production performance [2]. Contrarily, plant-sourced foods/proteins—mainly sourced from cereals and legumes and produced from conventional farming systems—are exposed to biotic (pathogens, pests), abiotic (climate variability and change and extreme weather event) and institutional (food trade restrictions) risk factors [3]. However, due to their intimate interaction with the environment, plant-sourced foods represent our best option for mitigating biotic and abiotic pressures and regenerating our natural resources. To achieve this, we need to move strategically from the green to gold agriculture revolution making novel, synthetic systems in crop plants (i.e., enhance the efficiency of photosynthesis systems by improving the carbon fixation reaction), which will boost agriculture production and secure food for the future generation. This improvement in natural resource usage efficiency [4,5,6] has the potential to deliver a step-change in agricultural output. On the other hand, environment-disconnected food systems have the potential to deliver risk-resilient diets but hardly directly address the climate challenges, biodiversity losses or support agriculture sectors and resilient landscapes.

Individuals in affluent societies consume more calories than they burn, partially caused by energy-rich food products, resulting in obesity and associated pathologies [7]. According to the World Health Organisation, non-communicable diseases (NCDs) are the leading cause of death (86%), disease (77%) and disability in Europe [8]. NCDs are largely preventable, and many initiatives are exploring prevention and control. The magnitude of western overconsumption of food surpasses that of food wasted in the household [9]. In this scenario, protein is significantly more critical than fats and carbohydrates, both numerically and environmentally, because the average highly digestible protein intake in many Western countries is 150–200% of the recommended value [10,11]. Protein overconsumption (i.e., protein that is nutritionally unnecessary) in western countries has been widely reported [11,12,13] and is far above the Population Reference Intake (PRI) [14]. There is a clear rationale to decrease the daily intake of protein since a substantial body of evidence associates the overconsumption of protein with adverse effects on human health, such as disorders of bone and calcium homeostasis, renal and liver dysfunction, increased cancer risk, hyperalbuminemia and precipitated progression of coronary artery disease [15,16,17,18,19,20]. Refs. [21,22], therefore, suggest a ‘reversed’ diet transition by ‘using less’ (e.g., leaving the meat out of the dish) or ‘doing things differently’ by a diet reformulation strategy, with reduced protein content in food products appears to be the most effective approach. However, plans to convince free and affluent societies to eat healthy but not innately desired food have been largely unsuccessful in the past [15,16,17,18,19,20,21,22]. Since the beginning of nutritional science, it has been hypothesised that the nutrients ingested through our diet are not entirely absorbed in the body, and only part of them are available. In such context, the terms “(bio)availability and (bio)accessibility” has come into use to identify such proportions [17]. The relatively recent recognition of incomplete protein digestion and absorption, mainly from vegetables, raises interest in non-digestible protein fractions [23]. These fractions may safely be called “resistant proteins” and are neither absorbed within the small intestine nor hydrolysable by mammalian digestive enzymes in the small intestine but may confer additional physiological benefits beyond the classical nutritive function of the protein. Despite the absence or presence of entanglements with nonprotein ingredients, some researchers [24,25,26,27,28] consider resistant proteins as proteinous dietary fibre and include them within the dietary fibre definition along with celluloses, hemicelluloses, lignins, oligosaccharides, pectins, gums and waxes, resistant starches and associated compounds such as polyphenols [29]. When incorporated into future foods, resistant proteins can impact other dietary components’ behaviours in food matrices, specifically carbo [30]. However, their technological potential and metabolic and physiological effects remain almost unstudied.

## 2. Resistant Plant Protein: Functions

Digestibility and amino acid composition have been recognised as essential factors for evaluating dietary protein quality [31]. For this reason, most legume proteins accumulated in seeds are still considered inferior in quality to animal protein, even though they have a physiological role far beyond the provision of essential amino acids with unexpected nutritional significance. In such regard, preliminary studies have shown how resistant proteins may exert physiological functions similar to dietary fibre as per se or through the interaction with other dietary constituents such as resistant starch by modulating its fermentation pattern in the large intestine with the increase of the short-chain fatty acid content [30] and the modulation of the gut microflora performance [30,32]. Besides containing small peptides and easily digestible proteins, legumes contain protein fractions that are either partially or entirely resistant to human digestive enzymes. These peptides and proteins may provide significant physiological and health-promoting effects, notably cholesterol-lowering activity [21,22], protecting cardiovascular health, reducing inflammation and cancer risk, weight control [33] and increased insulin sensitivity [34,35]. On the other hand, their structural stability has been reported to affect in vivo digestibility and availability of essential amino acids and the production of bioactive compounds. In addition, structural traits of legume proteins are of primary importance for their potential allergenicity and toxicity. These adverse effects must be carefully considered to exploit the beneficial effects of proteins and peptides from legume seeds [36]. Known classes of non-digestible bioactive legume protein and peptides are (i) storage proteins 7S and 11S, globulin, prolamin, glutenins from soybean and lupin with ACE-inhibitory properties, hypotensive, anticarcinogenic and anti-inflammatory activities [37,38,39]; (ii) lectins (carbohydrate-binding proteins) characterised by a tight β-sandwich structure that allows them to survive the acidic environment of the digestive tract where lectins exert anti-cytotoxic and anticancer activities [40]; (iii) glycated pea storage protein that is able, at least partially, to escape digestion and act as a modulator of the bacterial metabolic activities and their adhesive potentials [41]; (iv) α-amylase inhibitor from the white bean as an active agent in weight loss and glycaemic control [42]; and (v) protease inhibitor with anticarcinogenic activities [43]. Amylase and protease inhibitors are a heterogenous group of organic molecules, including proteins (>15 kDa) and peptides (<15 kDa), and are usually used by plants as defence strategies against pathogens, such as viruses, bacteria, or herbivores [44,45]

The compact structural feature of the protease inhibitor appears to have significant beneficial effects [46]. The main characteristics of the known undigestible proteins are depicted in Figure 1. Their structural peculiarity, interaction with other food constituents and low solubility are mainly responsible for their high stability and low digestibility [23].

Lectin, defensins, glycated protein and protein inhibitors have been widely investigated for their biological activities and represent the minor components of the non-digestible storage proteins/peptides. On the contrary, resistant protein, the principal constituent of the non-digestible storage protein, has been neglected. Therefore, its role in food design and human health has not yet been elucidated. Up to now, the full potential of resistant protein in food applications and human health enhancement remains untapped. Usually, plant-based resistant protein is separated during the industrial plant protein extraction process and discarded within the “fibre” side stream fraction. Over the past ten years, the scientific and industrial communities have focused on producing protein ingredients with high digestibility and optimal amino acid composition along with desired structure, rheology, palatability, flavour and appearance. However, much interest has recently been aroused in the new physiological function of these classes of proteins. From a physiological perspective, this new class of resistant protein can be proposed to be analogous to dietary fibre, potentially without the detrimental effects of some of the poorly absorbed fermentable oligo-, di-, monosaccharides and polyols (FODMAPs), highly present in the fibre fraction [47,48,49] and linked to irritable bowel syndrome (IBS) [50]. Moreover, the concept of digestible and indigestible proteins may be applied if a high concentration of amino acids in the plasma are detrimental to the patients—such as in metabolic genetic disorders (PKU), kidney deficiency or hepatic encephalopathy.

## 3. Resistant Plant Protein: Forms

The indigestibility trait of the resistant protein may originate from its structural peculiarities such as hydrophobicity, tertiary architecture characterised by a high content of β-sheet configuration, molecular conformation [51], the presence of thermally stable crosslinking formed by intra-and intermolecular hydrogen bonds and disulfide bridges [51] and its interaction with other food constituents such as carbohydrates [41,52]. Except for intrinsically occurring indigestible proteins, food processes applied during protein extraction/protein fractionation (acid or alkaline treatment) or food formulations (extrusion, boiling/cooking, fermentation) might build up indigestible protein species through aggregation, denaturation, polymerisation [51] and entanglement of proteins [23]. In investigating the effect of a highly resistant protein diet on young pig gut microbiomes, growth rate and metabolic profile, Murray and colleagues [53] manufactured a resistant protein diet by heating the feed (15 h at 70 °C followed by 20 min at 121 °C) to drive the Maillard chemistry of proteins and carbohydrates and confer digestive resistant status to the protein. The heat treatment of the resistant protein diet was designed to simulate the high heat processing that many ultra-processed food products undergo a concomitant development of resistant proteins. However, further investigation needs to be performed to provide robust evidence on such protein structure evolution. Similar to fibre components, resistant dietary proteins could have a disruptive effect on food structure by increasing matrix viscosity mainly due to their low water solubility, as previously reported for Marama bean proteins characterised by high β-sheet conformation hydrophobic interactions and tyrosine crosslinks [54]. Therefore, the inclusion of resistant protein in food formulation has to be adequately assessed regarding its potential structural interference with the food matrix architecture, rheology, colour, taste and appearance. However, much more investigation needs to be performed on these proteins’ physiological and nutritional significance in promoting the gut microbiota’s eubiosis condition that strongly influences our health and disease status.

## 4. Limitations

In this paper, the authors present and link preliminary data that need to be further validated with better animal models (e.g., growing pigs) or human clinical trials. Additionally, the fate of resistant protein passing into the colon requires an extensive investigation considering the positive and negative systemic and metabolic effects of colonic protein fermentation on the host [55]. In such regards, the resistant protein could reach the colonic microbiota and act as an amino acid source for protein fermenters, mainly species from *Clostridium*, *Desulfovibrio*, *Peptostreptococcus*, *Acidaminococcus*, *Veillonella*, *Propionibacterium*, *Bacillus*, *Bacteroides* and *Staphylococcus* [56,57]. In contrast to the extensively studied beneficial role of carbohydrate-derived short-chain fatty acids (SCFA), the effects of amino acid-derived SCFA on host physiology are not well known [58] and are associated with the production of other potentially harmful metabolites, including ammonia, sulfides and biogenic amines [59], among others with the potential capability to impact immunomodulatory, neurological, cardiovascular and gut functions [57,60,61]. These end-products may increase inflammatory response and tissue permeability and might be implicated in the development and severity of the symptoms of colorectal cancer and metabolic diseases, diabetes and non-alcoholic fatty liver disease [62]. A recent study conducted by Murray and colleagues [53] aimed to evaluate the effects of a standard vs. highly resistant protein diet on growth, gut microbiome, metabolomic profiles and the biomarkers of disease risk in pigs. The study demonstrates that the resistant protein was able to modulate the gut microbiome (metabolites) and negatively affect body mass and renal functions. Additionally, besides the potential health benefits of lectins, these sugar-binding proteins can bind to the surface of epithelial cells in the digestive system because of their high affinity for carbohydrates and can result in toxic reactions with changes in intestinal permeability [63,64]. In addition to the differences in protein digestibility due to protein source or processing factors, the variable capacities of individuals to lyse proteins (so-called digestive phenotype) may affect the abundance in which intact or partially degraded proteins are transferred to the large intestine [56].

## 5. Perspectives

The transition towards more inclusion of plant protein in our diets could bring us toward new and unexpected horizons regarding human health and wellbeing beyond the well-consolidated and known benefits. The investigation of the existence, distribution and physiological function of this class of proteinous dietary fibre could significantly contribute to longevity and public health, specifically in western countries where the increase in life expectancy is foreseen [40]. We suggest that the physiological significance of the resistant proteins, which are supposed to have only low nutritional value by the conventional nutritional assessment for protein, should be re-examined from this perspective (Figure 2).

Indeed, defining the new plant-resistant proteins and identifying their pre-and post-digestive multidisciplinary/interdisciplinary features will provide a crucial knowledge basis that will:-Open a completely new research field on resistant protein, combining the interests of food scientists and food engineers to develop strategies for designing future foods.-Advance the discipline of nutritional science by delivering comprehensive investigations elucidating the role of resistant protein in the maintenance of health and the prevention of non-communicable disease. The scientific advancement in this research field would significantly impact the future of food nutrition, public health and food policy development.-Allow the food industry, in conjunction with national and international health authorities, to implement health promotion dietary strategies (personalised ingredients/foods) by establishing a pipeline in which the characterised resistant plant protein will be linked to NCD prevention for the potential as diet-based therapeutics.-Endorse and facilitate a dietary shift to include resistant plant protein by consumers seeking a healthier, nutritious and more sustainable diet.

Every new finding brings new questions. For example, can we expect to identify these dietary proteins in other food sources such as macro/microalgae, insects and fungi? Are there striking differences among resistant proteins from different food sources? How do these protein categories modulate the physiological behaviour of the other food constituents in the large intestine? Can these proteins be ex situ synthesised? Can we use these proteins to design the future of food according to the citizen’s health requirements? Perhaps the first step is to elucidate further the physicochemical and biological peculiarities that characterised these new proteinous dietary fibre constituents.

## Figures and Tables

**Figure 1 foods-11-02759-f001:**
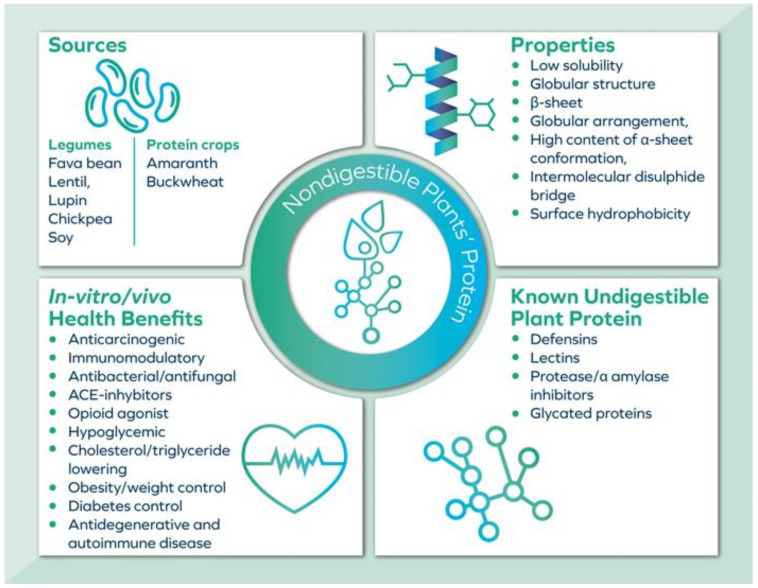
Principal physicochemical and biological properties of known classes of indigestible plant protein.

**Figure 2 foods-11-02759-f002:**
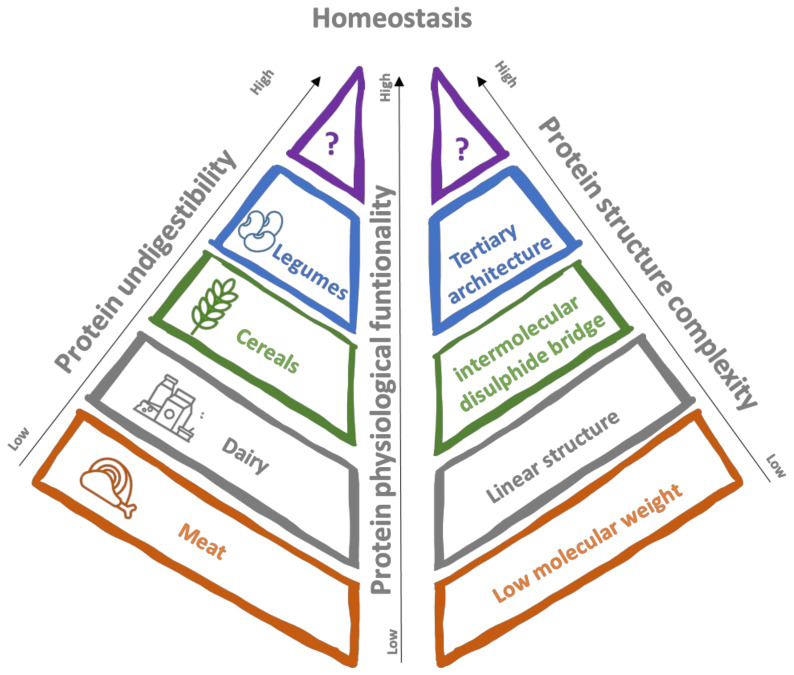
Overall representation of the proteinous fibre concept. The degree of protein indigestibility is strictly influenced by the raw materials where they are sourced and by the different food (bio)processing they undergo during their extraction, purification and subsequent inclusion in food products. However, learned scientist participation in this immature field is eagerly awaited.

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
