# Peer review of "Resistant Protein: Forms and Functions"

_foods, 2022, doi:10.3390/foods11182759_

Round 1

Reviewer 1 Report

·        L8 – Many ….eat – Please consider a change to another form of introduction.

·        L11 – Is it various potential toxic end products? I think a comma is not necessary after toxic.

·        L18-20 – rewrite this sentence.

·        L20-22 – How is this information relevant to this review?

·        L23 – please add more keywords

·        L26-30 -This is not direct relevance to this review, and the authors mentioned that addressing this issue need protein production, but protein is not just food alone; there are other forms of nutrients, which completely replace protein, still in any part of the world, taking protein, not as a main meal. Please consider revising this point.

·        L46-47 – While a healthy…… remain very confusing, please rewrite.

·        I recommend that authors remove L46-59 and replace it with more relevant details linked to protein diet-based problems to health and why the resistant protein is in need etc.

·        L60-63- author recommends decreasing the protein intake, but based on what ground? Just with single reference and some listed diseases may not be enough to convince the readers. Please provide a clear picture of it.

·        What are the main health benefits of resistant proteins? What happens to the body if consumes a high quantity of resistant proteins? Will it cause an adverse effect?

·        Generally, certain forms of protein trigger inflammations and various allergic reactions; the author should address this point in the review.

·        Is the resistant protein can be externally manufactured? Alternatively, please include a section and discuss how to isolate them from the plant source.

·        Amylase and protease inhibitors, what are those, have any specific chemical structure or name? Please provide more details.

·        L145-148 – All those listed methods could hydrolyze the protein in whole or partially; will it still be a resistant protein?

·        If possible, the author provides an infographic of resistant protein mode of action in controlling a chronic disease, which would greatly strengthen this review.

·        L159 – consider removing this section if the provided data has not valid.

·   Author is also promoting proteinous fibre, but not used much in the food use it, but also author did not provide more information on this term of usage. 

Reviewer 2 Report

This short review on resistant protein highlights an important dietary constituent that has perhaps not received proper attention when considering the strong focus on resistant starches.  Food products that escape digestion and move into the colon are fermented to various products  – often promoting healthful outcomes.  The authors briefly mention in the abstract the ‘potentially toxic, end products’ of resistant protein fermentation – but this is not discussed in the review itself.  The authors need to expand the text to include a section specifically outlining the products of resistant protein fermentation (e.g., peptides and amino acids as stated in the abstract) and how they may impact health.  The text at the top of page 3 seems to be outlining potential benefits, but this discussion is very vague and not link to any fermentation products.  There is no mention in the review of the ‘toxic’ end products stated in the abstract – which is frustrating for a reader.  For example, in pigs, feeding resistant protein resulted in decreased growth and an elevated kidney disease risk.  This literature needs to be discussed. 

Line 42:  provide the reader with more background on ‘gold’ agriculture.  Be sure to provide the reader with several key citations on the topic.

Lines 46-59: this text is not well referenced.  For example, a citation is not provided for the text stating ‘according to the WHO….’ 

Line 57:  the discussion regarding ‘reduced protein content in food products’ suggests this is a necessity.  Yet 50% of children in developing countries have a degree of protein malnourishment.   In the U.S. 8% of American woman consume below the recommended amount for protein.  Low protein intakes are also an important concern for the elderly.  The population that is over consuming protein are adult men in the U.S.  It seems that the most ‘effective approach’ to reduce protein intake in the U.S. is to lower the amount consumed – not produce food with more resistant protein.  Or perhaps substitute meat dishes with legume dishes.  Incorporating resistant proteins into ‘future foods’ seems to create extra steps, steps that might adversely affect certain population groups who consume too little protein.  The authors should consider the ‘unintended consequences’ of this concept. 

Line 52:  the ‘overconsumption’ of food is linked to high intakes of fat and carbohydrate as well.  How will increasing resistant proteins in foods help this this issue?

Line 63:  cite a meta-analysis showing the actual data and strength of association

Line 66:  the term ‘results’ is not appropriate.  You are referring to the monosaccharides, fatty acids, and amino acids, right?

Line 67:  ‘recent recognition’ refers to a 22-year-old paper.

Line 78:  There is no discussion in this review on the interaction of resistant protein on carbohydrates – this is the ‘missing section’ discussed at the start of this critique. 

Line 92:  reference is needed

Lectins (line 104) are ‘anti-nutrients’ – they need to be described further in the ‘missing section’ discussed above.

Line 122:  define ‘non-digestible storage protein’ --  it seems that this concept be introduced earlier in the review. 

Round 2

Reviewer 1 Report

All the minor points are addressed. 

However, I have requested several information and inclusion of details in this review but authors have shown no interest in including those points, which I think necessary for further consideration of this article. 

So I include the review comments below here again that definitely need into include in this review, please see the followings:

Q: I recommend that authors remove L46-59 and replace it with more relevant details linked to protein diet-based problems to health and why the resistant protein is in need etc.

 Q: Generally, certain forms of protein trigger inflammations and various allergic reactions; the author should address this point in the review. [I could not able to track the information referred in the L213-235.

Q: Is the resistant protein can be externally manufactured? Alternatively, please include a section and discuss how to isolate them from the plant source.

"Amylase and protease inhibitors are an heterogenous group of organic 125 molecules including proteins (>15 kDa) as well as peptides (<15 kDa), and usually are 126 used by plants as defense strategies against pathogens, such as viruses, bacteria, or 127 herbivores [45]. " - Should include the mode of action of how these enzymes defense against the pathogens? normally these are hydrolytic enzymes to break macromolecules, and the microogranism or any other organisms when its in tack it cant be been broken down, if other mentioning defense action, then please include some more informations.

Q: L145-148 – All those listed methods could hydrolyse the protein in whole or partially; will it still be a resistant protein? Where is this details, is authors removed the table or the information from the paper?

Q:If possible, the author provides an infographic of resistant protein mode of action in controlling a chronic disease, which would greatly strengthen this review

Q: L159 – consider removing this section if the provided data has not valid.

I recommend author's to address this points so that I can provide my final recommendation to the journal, at present, it is is not fulfilled, and furthermore, provide highlights for your revision, as the tracking too hard to find what exactly you have revised, and the referring number in the comments and revision in the paper is not matching, please check it carefully.

Thanks

Reviewer 2 Report

All the comments were adequately addressed. 

Author Response

We thank the referee.